# Noise as a Knob: An Inference Time Noise Scheduling Strategy to Optimize Diffusion Based Vision Tasks

## Abstract

Diffusion models, originally developed for generative tasks, are increasingly showing promise in discriminative vision tasks like segmentation. Several studies have showcased their adaptability, with diffusion-based generalized frameworks simplifying complex architectures by unifying various components. Despite their architectural elegance, these models often face performance gaps when compared to established GAN and transformer-based methods. This paper delves into the limitations of diffusion models, particularly observing their tendency to prioritize recall over precision. To address this, we introduce a novel inference-time noise scheduling strategy that dynamically adjusts noise during the reverse diffusion process. Crucially, this method requires no additional training of the diffusion model. Our strategy significantly enhances precision with minimal recall reduction for pre-trained models. This leads to an improved Panoptic Quality (PQ) of 52.7 on the COCO validation dataset. While still trailing top performing transformer-based methods, our approach improves the panoptic segmentation benchmark among generalized diffusion-based frameworks by 1.5%. We also show our approach enhances panoptic segmentation in adverse weather. Furthermore, we validate its versatility in text-to-image generation, achieving an X-IQE image-text alignment score of 4.6 on DrawBench, improving the baseline score of 3.6. Our method provides a flexible and effective tool for optimizing task-specific performance and enhancing the utility of diffusion models across both generative and discriminative applications, all without requiring retraining.

## 1 Introduction

Diffusion-based models have emerged as a powerful paradigm in computer vision, achieving state-of-the-art performance in generative tasks such as image synthesis, inpainting, and super-resolution (Rombach et al., 2022; Dhariwal & Nichol, 2021; Saharia et al., 2022b). Their ability to model complex data distributions has recently sparked interest in applying diffusion models to dense prediction tasks, including depth estimation (He et al., 2024; Ke et al., 2024), image segmentation (Tan et al., 2022; Qiu et al., 2024), object detection (Chen et al., 2023a; Wang et al., 2024), and panoptic segmentation (Van Gansbeke & De Brabandere, 2024; Xu et al., 2023). However, despite their promise, diffusion-based approaches still lag behind discriminative models in terms of overall performance on dense prediction benchmarks.

For instance, the generalized diffusion-based panoptic segmentation framework currently achieves a Panoptic Quality (PQ) of 51.9 on the COCO validation dataset. This is significantly lower than Mask DINO, a state-of-the-art transformer-based model that achieves a PQ of 59.5 (Li et al., 2023). Similarly, in semantic segmentation, the best diffusion-based model achieves an mIoU of 52.8 (Tan et al., 2022), whereas transformer-based models like BEiT (Wang et al., 2023b) reach an mIoU of 62.8. These performance gaps raise an important question: Why do diffusion models lag behind their transformer-based counterparts in dense prediction tasks, and how can we close this gap?

A fundamental difference between transformer-based and diffusion-based dense predictors lies in their underlying modeling approach. Transformer-based models, such as Mask DINO, are discriminative, they directly map an input image to its segmentation labels using a DETR-style transformer

decoder, without explicitly modeling the data distribution. Diffusion-based models, in contrast, are generative in nature, they learn the entire data distribution and generate dense predictions by iteratively refining a noisy input. This distinction has key implications for performance:

- Diffusion models inherently prioritize recall (Table 1), as they explore diverse possible segmentations by modeling the underlying data distribution.
- However, this comes at the cost of precision, as diffusion models can introduce spurious predictions, leading to lower Panoptic Quality compared to discriminative methods.

In this paper, we systematically investigate the cause of low precision in diffusion-based panoptic segmentation. Our key observation is that while the diffusion model's recall is naturally high, its precision is limited due to the inherent stochasticity in the reverse process. For example, Mask2Former (Cheng et al., 2022), a non-diffusion transformer model for panoptic segmentation, has a precision value of 80.4 compared to our diffusion baseline (Van Gansbeke & De Brabandere, 2024) which has a precision of 76.1. To address this shortcoming, we propose a novel inference-time noise scheduling strategy that biases the diffusion process towards more deterministic and structured predictions, thereby improving precision while maintaining recall. With noise fine-tuning, our framework achieves 1.5% improvement in Panoptic Quality over the current state-of-the-art-method among generalized frameworks on COCO Lin et al. (2014) dataset. We validate the method's sensitivity by testing PQ on unseen ADE20K dataset 6 and also experimenting with different levels of synthetic image degradation on the COCO validation dataset5.3.

Our scheduling approach also demonstrates broader applicability across diverse generative tasks. For text-to-image generation on the DrawBench (Saharia et al., 2022a) dataset, we achieve a significantly improved explainable Image Quality Evaluation (X-IQE) (Chen et al., 2023c) Image-Text Alignment score of 4.6, compared to the Stable Diffusion 1.5 baseline of 3.6. This further validates that our new strategy leads to more deterministic results, suggesting similar benefits for other complex vision problems.

Our contributions are threefold:

- We demonstrate that diffusion-based segmentation is inherently recall-centric.
- We propose a novel inference-time noise scheduling strategy that improves the precision of diffusion models for panoptic segmentation.
- We show the wider applicability of using the noise schedule as an inference-time knob to optimize pre-trained diffusion models for both generative and dense prediction tasks.

## 2 RELATED WORK

### 2.1 CONDITIONAL DIFFUSION MODELS

Conditional diffusion models are widely employed for image generation and dense prediction tasks. For image generation, conditioning is typically provided by a text prompt (Ramesh et al., 2022; Zhang et al., 2023). In dense prediction tasks, the conditioning takes the form of an input image for which the dense prediction is required. This conditioning guides the model to generate the relevant output. The diffusion model consists of a forward process for training and a reverse process that is utilized for inference.

The forward diffusion process is defined as follows:

$$q(x_t|x_{t-1}, c) = \mathcal{N}\left(x_t; \sqrt{\alpha_t}\, x_{t-1} + f(c),\, (1 - \alpha_t)I\right), \tag{1}$$

where $\alpha_t$ is the variance scheduling term, and $f(c)$ is a function (e.g., implemented via cross-attention or concatenation) that injects the conditioning information (such as segmentation cues) into the process using the conditioning latent $c$.

A noisy sample at any timestep $t$ can be expressed as:

$$x_t = \sqrt{\bar{\alpha}_t}\, x_0 + \sqrt{1 - \bar{\alpha}_t}\, \epsilon, \quad \epsilon \sim \mathcal{N}(0, I), \tag{2}$$

with the cumulative product defined as $\bar{\alpha}_t = \prod_{s=1}^{t} \alpha_s$.

The reverse process recovers $x_0$ from $x_t$ by iterative denoising while being conditioned on $c$:

$$p_\theta(x_{t-1} \mid x_t, c) = \mathcal{N}(x_{t-1}; \mu_\theta(x_t, t, c), \Sigma_\theta(x_t, t, c)), \tag{3}$$

where $\epsilon_\theta(x_t, t, c)$ is a denoising network that predicts the added noise given $x_t$ and condition $c$. The predicted noise is then used to compute the mean $\mu_\theta(x_t, t, c)$ according to the DDPM parameterization, while $\Sigma_\theta(x_t, t, c)$ denotes the (fixed or learned) variance of the reverse process.

## 2.2 PANOPTIC SEGMENTATION

Panoptic Segmentation (Kirillov et al., 2019) combines semantic and instance segmentation. It involves assigning each pixel a semantic label for things like car, grass etc. as well as an instance id to differentiate between different instances of the same class. This creates a comprehensive understanding of the image required for tasks like autonomous navigation, robotics and satellite imaging. Panoptic Segmentation is measured using the Panoptic Quality(PQ) metric defined as follows:

$$PQ = \frac{\sum\limits_{(p,q)\in TP} IoU(p,q)}{|TP| + \frac{1}{2}|FP| + \frac{1}{2}|FN|} \tag{4}$$

where, True Positives (TP) is the set of correctly matched predicted $p$ and ground-truth $q$ segments with $IoU > 0.5$. False Positives (FP) are predicted segments that do not match any ground-truth segment. False Negatives (FN) are ground-truth segments that do not match any prediction. $IoU(p,q)$ is the Intersection-over-Union between matched pairs.

There are several research works on panoptic segmentation. Many of them are extensions of previous research on segmentation and object detection. Panoptic segformer (Li et al., 2022) uses multi-level feature aggregation, and a query-based instance segmentation head to extend segformer (Xie et al., 2021) semantic segmentation architecture for panoptic segmentation. Mask DINO (Li et al., 2023) extends the DINO (Caron et al., 2021) object detection framework using an additional mask prediction branch for panoptic segmentation. Mask2Former (Cheng et al., 2022) proposes a masked attention based transformer architecture where the cross-attention is constrained by a mask for a generalized segmentation framework. Chen et al. (2023b) propose a method to perform panoptic segmentation on both images and videos using a transformer based architecture.

## 3 NOISE AS A KNOB

Diffusion models are designed to learn the underlying distribution of training data and generate panoptic segmentation labels by progressively denoising a noisy input. These models typically rely on a conditional image input to guide the denoising process toward semantically meaningful outputs. In contrast, transformer-based models process an input image in a fully discriminative manner, directly predicting segmentation labels without the need for iterative refinement. This often results in deterministic outputs. A fundamental distinction between these two paradigms lies in their inference mechanisms. Diffusion models employ a stochastic sampling process, introducing variability in their predictions, which can enhance diversity but may also lead to inconsistencies. In contrast, transformer-based architectures generate segmentation outputs in a single forward pass, ensuring deterministic and stable predictions.

### 3.1 MODIFYING THE DIFFUSION REVERSE PROCESS

Prior work (Singh et al., 2022; Ahn et al., 2024; Ho & Salimans, 2022) shows that increasing the noise level during sampling encourages the model to rely more heavily on conditioning signals. In parallel, Kingma et al. (2021) demonstrate that the variational lower bound of diffusion models is largely invariant to small perturbations of the noise schedule, except through the signal-to-noise ratio (SNR) near the trajectory endpoints. Motivated by these findings, we design an inference-time modification that raises the noise levels in the mid-trajectory while keeping the endpoints close to the training schedule. This requires no retraining and biases the model toward more deterministic, condition-driven reconstructions.

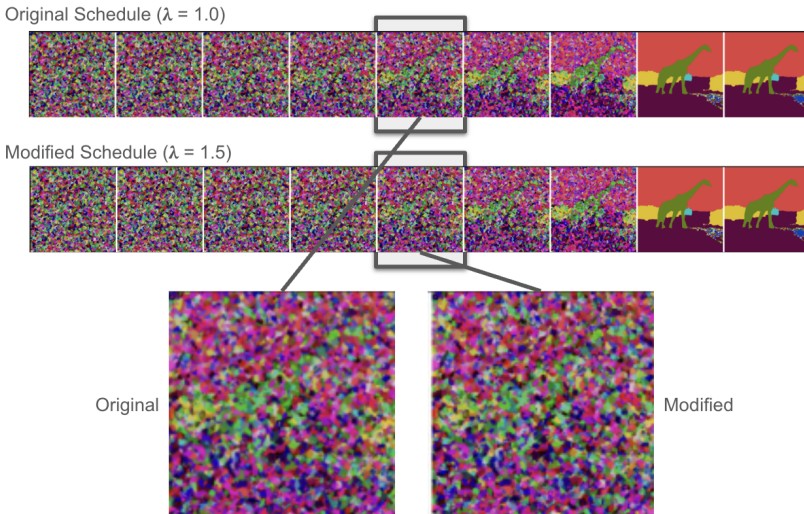

Figure 1: Impact of $\lambda$ (equation 6) on the Noise Schedule of diffusion process. It can be observed that the modified noise schedule has more noise and less signal, compared to the original schedule. In the original schedule, structural details such as the giraffe outline remain partially visible due to a higher signal-to-noise ratio at intermediate timesteps. In contrast, the modified schedule introduces more noise, suppressing these high SNR features and making the conditional input more influential in guiding the denoising process. The difference is particularly evident in the zoomed-in images, where background structures(e.g. the yellow region) are less discernible under the modified schedule.

At each reverse step, the clean sample is reconstructed from the noisy state $x_t$ as

$$\hat{x}_0(x_t, t, c) = \frac{x_t - \sqrt{1 - \bar{\alpha}_t}\, \epsilon_\theta(x_t, t, c)}{\sqrt{\bar{\alpha}_t}}, \tag{5}$$

where $\epsilon_\theta(x_t, t, c)$ is the condition-aware noise prediction. We modify this step by replacing $\bar{\alpha}_t$ with $\bar{\alpha}_t^\lambda$:

$$\hat{x}_0^{(\lambda)}(x_t, t, c) = \frac{x_t}{\sqrt{\bar{\alpha}_t^\lambda}} - \frac{\sqrt{1 - \bar{\alpha}_t^\lambda}}{\sqrt{\bar{\alpha}_t^\lambda}}\, \epsilon_\theta(x_t, t, c). \tag{6}$$

**Effect on SNR:** Substituting Eq. 2 into Eq. 5, the reconstruction error becomes

$$\hat{x}_0 - x_0 = \frac{\sqrt{1 - \bar{\alpha}_t}}{\sqrt{\bar{\alpha}_t}} (\epsilon - \epsilon_\theta). \tag{7}$$

Hence, the effective signal-to-noise ratio is

$$\mathrm{SNR}_t = \frac{\mathrm{Var}[x_0]}{\mathrm{Var}\left[\frac{\sqrt{1 - \bar{\alpha}_t}}{\sqrt{\bar{\alpha}_t}}(\epsilon - \epsilon_\theta)\right]} \quad \propto \quad \frac{\bar{\alpha}_t}{1 - \bar{\alpha}_t}. \tag{8}$$

Our modification (Eq. 6) leads to a modified SNR as follows,

$$\mathrm{SNR}_t^{(\lambda)} \quad \propto \quad \frac{\bar{\alpha}_t^\lambda}{1 - \bar{\alpha}_t^\lambda}. \tag{9}$$

For $\lambda > 1$, we have $\bar{\alpha}_t^\lambda < \bar{\alpha}_t$, and therefore

$$\mathrm{SNR}_t^{(\lambda)} < \mathrm{SNR}_t. \tag{10}$$

This reduced SNR amplifies the role of the model's noise prediction $\epsilon_\theta(x_t, t, c)$. As $\bar{\alpha}_t^\lambda$ decreases, the scaling factor $\frac{\sqrt{1 - \bar{\alpha}_t^\lambda}}{\sqrt{\bar{\alpha}_t^\lambda}}$ (Eq. 6) grows, making the reconstruction increasingly dependent on

correcting the injected noise. This forces the reverse process to rely more strongly on condition-aligned predictions, yielding reconstructions with higher precision but reduced diversity. Note that towards the denoising trajectory endpoints $\bar{\alpha}_t$ is close to either 0 or 1 so the scaling by $\lambda$ has minimal impact on its value. This aligns with the analysis of Kingma et al. (2021) and keeps the SNR at endpoints similar to the original schedule.

## 3.2 ANALOGY TO CLASSIFIER-FREE GUIDANCE (CFG)

This mechanism is analogous to the widely known technique of Classifier Free Guidance (CFG). CFG strengthens the influence of the conditional input (like a text prompt or an image) by pushing the prediction away from an unconditional noise estimate and further towards the conditional one. The effect of the scaling parameter $\lambda$ can be seen as an implicit guidance scale. In standard CFG, a guidance scale explicitly pushes the output to be more relevant to the condition. In Noise as a Knob, increasing $\lambda > 1$ implicitly amplifies the influence of the conditional signal $c$ that is embedded in the network's prediction $\epsilon_\theta(x_t,\ t, c)$. By re-weighting the reconstruction formula, the method forces each denoising step to rely more heavily on the learned, condition-specific part of its prediction. This stronger guidance leads to more deterministic and structured outputs that are better aligned with the conditional input.

## 4 EXPERIMENT SETUP

We test our approach with the panoptic segmentation architecture called LDMSeg (Van Gansbeke & De Brabandere, 2024). It is a generalized framework for dense prediction developed on top of the popular stable diffusion architecture. The latent space size is $[B, 4, 512, 512]$ where B is the batch size. The output has 128 channels to reconstruct 128 instances in a scene. The model was trained using the COCO (Common Objects in Context) (Lin et al., 2014) training dataset. COCO dataset is a widely used for tasks like object detection and panoptic segmentation. It has 80 object categories that include humans, animals, household items, vehicles etc. The training set has 118000 images and validation consists of 5000 images with annotated labels for panoptic segmentation.

We utilize the pre-trained model provided in LDMSeg and modify its inference noise schedule with varying $\lambda$ for our experiments. We do not conduct any kind of training or fine-tuning. The main focus for these tests is to measure precision, recall and Panoptic Quality. Note that it is a standard practice in previous panoptic segmentation research to evaluate performance using a class agnostic panoptic quality metric on the coco validation dataset. Most of the research works including our baseline approaches (Van Gansbeke & De Brabandere, 2024; Wang et al., 2023b;a) also do the same. Our results are shown is Table 1.

For simulating Haze on COCO validation set, we utilize the method described in (Agarwal et al., 2025; Liu et al., 2022). The hazy images can be synthesized using the atmospheric scattering model (Cantor, 1978; Narasimhan & Nayar, 2002) as follows:

$$J(x) = I(x)t(x) + A(1 - t(x)), \tag{11}$$

where, $J(x)$ is the hazy image, $I(x)$ is the original clean image, $A$ is the global atmospheric light whose value is set to 0.5, and $t(x)$ is the medium transmission map. The transmission map is computed based on the atmospheric scattering coefficient $\beta$ and scene depth $d(x)$:

$$t(x) = e^{-\beta}d(x) \tag{12}$$

$$d(x) = -0.04\rho + \sqrt{max(rows, cols)} \tag{13}$$

where $\rho$ denotes the Euclidean distance of the pixel from the image center, and rows and cols correspond to the dimensions of the image. Atmospheric scattering coefficient $\beta$ can be varied to obtain different levels of haze. We add 3 different levels of haze on the COCO dataset with $\beta$ values of 0.05, 0.10 and 0.15 and compute the performance. All our experiments used a slurm cluster node with 8 Nvidia A100 GPUs with memory of 80GB. The batch size was set to 32.

To demonstrate the generalization of our approach to generative tasks, we applied the inference schedule modification to the text-to-image generation using Stable Diffusion 1.5 (SD1.5) on Draw-Bench (Saharia et al., 2022a) dataset. Note that SD1.5 is known for stronger text-image alignment compared to newer versions. DrawBench is a challenging dataset specifically designed to evaluate text-to-image diffusion models, introduced by Google Research. It consists of 200 prompts designed to test a model's ability to understand and generate complex textual instructions. We compare the different schedules using explainable image quality evaluation (X-IQE) (Chen et al., 2023c) which uses MiniGPT-4 to evaluate text-to-image models on Fidelity, Alignment and Aesthetics.

## 5 RESULTS

### 5.1 PRECISION RECALL STATISTICS WITH THE MODIFIED NOISE SCHEDULE

Table 1: Comparison of precision, recall, and panoptic quality on the COCO validation dataset at different values of $\lambda$. (*) $\lambda = 1.0$ represents the baseline using the original training noise schedule, showing the highest recall. For $\lambda$ values greater than 1.0, precision consistently increases while recall experiences only a small reduction. Note the highest precision is observed at $\lambda = 1.5$. The optimal Panoptic Quality (PQ) of **52.7** is achieved at $\lambda = 1.3$.

| $\lambda$ | **Precision** | **Recall** | **PQ** |
|---|---|---|---|
| 0.8 | 70.6 | 52.8 | 49.3 |
| 0.9 | 74.2 | 53.8 | 51.1 |
| 1.0* | 76.1 | **54.2** | 51.9 |
| 1.1 | 77.4 | 54.2 | 52.4 |
| 1.2 | 78.3 | 54.1 | 52.6 |
| **1.3** | 78.9 | 53.8 | **52.7** (+1.5%) |
| 1.4 | 79.1 | 53.4 | 52.5 |
| 1.5 | **79.8** (+4.9%) | 52.9 | 52.5 |

We first conduct experiments with COCO validation dataset with the modified diffusion inference de-noising schedule. We test the effect of different values of $\lambda$ on precision, recall and panoptic quality metric. The results are shown in Table 1. Having $\lambda$ value less than 1 (0.8 and 0.9), leads to reduction in both precision and recall as well as have a lower panoptic quality score than the original schedule. For $\lambda$ values greater than 1, there is improvement in the precision score with slightly lower recall. The best panoptic quality of 52.7 is obtained at $\lambda = 1.3$. If we look at $\lambda$ values more than 1.3, the precision increases further but the recall reduces and it leads to lower panoptic quality.

### 5.2 PANOPTIC QUALITY PERFORMANCE COMPARISON

Table 2: This table presents the comparison of our approach with the SOTA methods on Panoptic Quality(PQ) metric on COCO validation dataset for class agnostic panoptic segmentation. We can see that our modified inference noise schedule based method outperforms all the generalist approaches. It sets a new benchmark for diffusion model based Panoptic Segmentation.

| **Framework** | **Backbone** | **#Params** | **Image Size** | **PQ** |
|---|---|---|---|---|
| *Specialist Approches:* | | | | |
| Mask2Former | Swin-L | 216M | 1024x1024 | 57.8 |
| kMax-DeepLab | ConvNeXt | 232M | 1281x1281 | 58.1 |
| Mask Dino | Swin-L | 223M | 1024x1024 | **59.4** |
| *Generalist Approches:* | | | | |
| Painter | ViT-Large | 370M | 448x448 | 41.3 |
| UViM | ViT-Large | 939M | 1280x1280 | 45.8 |
| Pix2Seq-D | ResNet | 94.5M | 1024x1024 | 50.3 |
| LDMSeg | UNet (Diffusion) | 851M | 512x512 | 51.9 |
| **Ours** | UNet (Diffusion) | 851M | 512x512 | **52.7** |

We evaluate the effectiveness of our modified inference noise schedule by comparing it against state-of-the-art (SOTA) methods on the COCO validation dataset. The quantitative results are summarized in Table 2. Our proposed approach, utilizing a rescaled inference noise schedule with a scaling factor $\lambda = 1.3$, achieves a Panoptic Quality (PQ) score of 52.7, surpassing existing generalist methods. This result highlights the effectiveness of our inference-time optimization in improving segmentation performance without requiring additional training. However, as expected, specialized task-specific models continue to outperform generalist approaches, benefiting from domain-specific optimizations and additional training strategies.

### 5.3 SENSITIVITY TO DIFFERENT LEVELS OF HAZY DEGRADATION

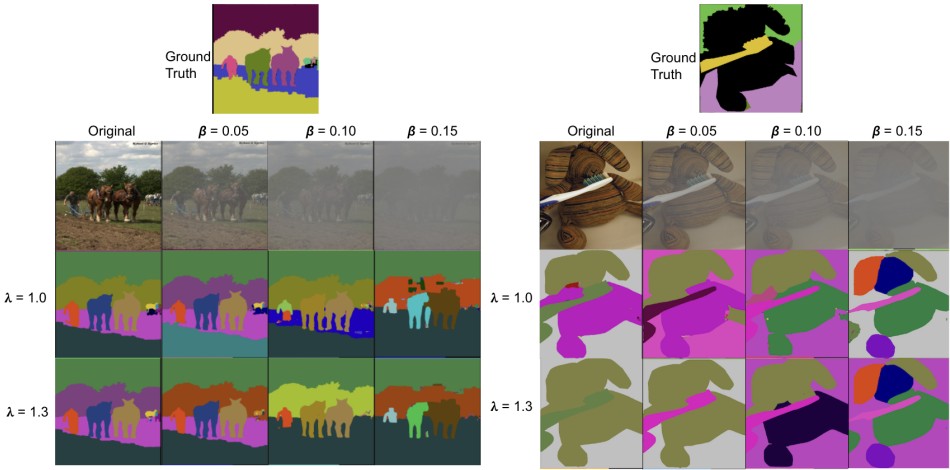

Figure 2: Comparison of segmentation quality using the baseline noise schedule ($\lambda = 1.0$) and the modified schedule ($\lambda = 1.3$) across varying haze levels $\beta$ on COCO validation images. Segment color assignment is arbitrary. Notice that the baseline schedule tends to over-segment, introducing spurious regions evident in the horse image at $\beta = 0.15$ and in all teddy bear examples, where parts appear in different colors. The modified schedule reduces these artifacts.

Table 3: Comparison of Panoptic Quality(PQ) on COCO simulated hazy dataset with different levels of haze ($\beta$). (*) The original training noise schedule is with $\lambda = 1.0$. Tweaking $\lambda$ improves PQ at different degradation levels.

| $\lambda$ | **Original** | $\beta = 0.05$ | $\beta = 0.10$ | $\beta = 0.15$ |
|---|---|---|---|---|
| 1.0* | 51.9 | 51.0 | 48.3 | 41.8 |
| 1.1 | 52.4 | 51.4 | 48.5 | **42.0** |
| 1.2 | 52.6 | **51.6** | **48.6** | 41.8 |
| 1.3 | **52.7** | 51.5 | 48.6 | 41.6 |
| 1.4 | 52.5 | 51.4 | 48.3 | 41.1 |
| 1.5 | 52.5 | 51.3 | 48.0 | 40.2 |

From our experiments it is clear that our modified inference noise schedule improves the precision value compared to the baseline. In this experiment, we see how the optimized schedule perform on different levels on hazy degradation compared to the training schedule. Comparison of Panoptic Quality on different levels on haze is shown in Table 5.3. We can still see improvement in terms of Panoptic Quality with modified schedule on different levels of haze. For both $\beta = 0.05$ and $\beta = 0.10$, the highest PQ is obtained with $\lambda = 1.2$, and $\lambda = 1.3$, also providing a boost in performance for these degradations. At $\beta = 0.15$ which is the highest level of degradation, the best PQ is obtained for $\lambda = 1.1$. It can be inferred that as the degradation level increases, the effect of scaling becomes less prominent and optimum value of $\lambda$ gets closer and closer to the baseline PQ. This is substantiated by the fact that at higher degradation levels, objects are often blurred so a high precision system will discard many detections. In this case, we should rely on a high recall system.

Figure 2 shows segmentation results with modified noise schedule across haze levels ($\beta$). As degradation increases, segmentation quality declines in both examples. The baseline schedule ($\lambda = 1$) tends to over-segment, introducing artifacts, clearly visible in the second teddy bear image and the horse image at $\beta = 0.15$. This aligns with its higher recall at the cost of precision. In contrast, the modified schedule ($\lambda = 1.3$) improves precision, producing smoother masks. This trade-off can be leveraged for task-specific needs.

## 5.4 RESULTS WITH TEXT TO IMAGE

Table 4: Text-to-Image performance comparison with Stable Diffusion 1.5 on DrawBench Dataset with modified inference-time noise schedule. We use **X-IQE** evaluation (Chen et al., 2023c) metrics. Percentage increase/decrease is calculated relative to the $\lambda = 1.0$ baseline.

| $\lambda$ | Fidelity | Alignment | Aesthetics | Overall |
|---|---|---|---|---|
| 1.0 | 4.7 | 3.6 | **3.9** | 12.2 |
| 1.1 | 5.4 (+16.0%) | 4.5 (+25.8%) | 3.1 (-20.8%) | 13.1 (+7.1%) |
| 1.2 | **5.5** (+17.7%) | **4.6** (+28.3%) | 3.3 (-14.4%) | **13.5** (+10.6%) |

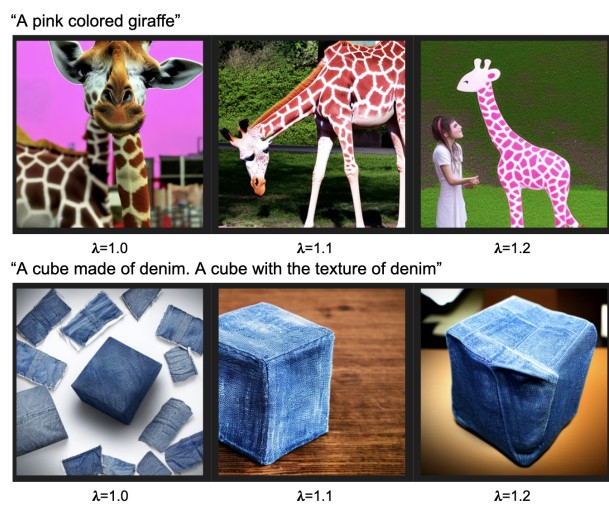

Figure 3: Sample images with improved image text matching generated from the DrawBench dataset prompt (given above) with varying inference noise schedule.

To measure the generalization of using this inference time scheduling, we also tested with the modified noise schedule on DrawBench Dataset. The results are presented in Table 4. It shows comparison of Fidelity, Image-Text Alignment, Aesthetics and overall X-IQE (eXplainable Image Quality Evaluation) score. We see that the Alignment score increases with increasing $\lambda$, substantiating our panoptic segmentation results and showing that having more noise in the schedule makes the conditional input more relevant. This comes at the cost of aesthetics.

When we increase $\lambda$ to 1.1 the Alignment score increases from 3.6 to 4.5 and $\lambda = 1.2$ leads to even higher score of 4.6. Overall Image Quality also improves from 12.2 to 13.5. Figure 3 shows a few qualitative examples where modified schedule improves the image-text matching. Beyond $\lambda = 1.2$ the Aesthetics score drops a lot. Its also visible in the Giraffe image where the Giraffe is now pink aligning with the prompt but the image looks more cartoonish.

## 5.5 TESTING WITH ALTERNATIVE NOISE SCHEDULES

In this paper we focused on a specific variation of noise schedule where $\bar{\alpha}$ is scaled by power of $\lambda$. We complied with the conditions suggested by Kingma et al. (2021) which says that the inference

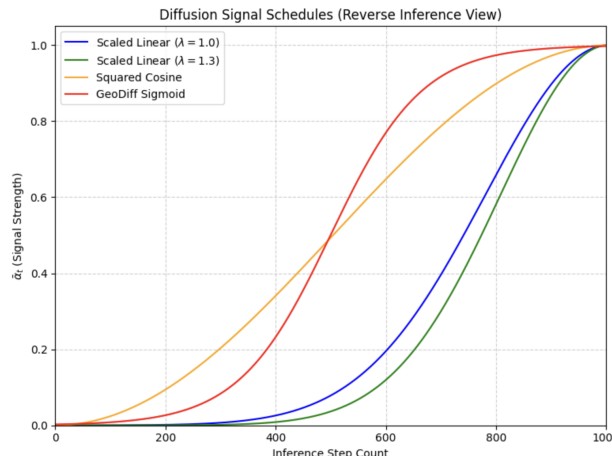

Figure 4: Comparison of $\bar{\alpha}$ values for different noise scheduling strategies. The baseline is the blue curve representing the scaled linear schedule. We can see that Squared cosine (Nichol & Dhariwal, 2021) and GeoDiff sigmoid (Hoogeboom et al., 2022) strategies deviate from the baseline.

time noise schedule in the middle of the diffusion process can be modified without breaking the diffusion framework. We also did experiments with other common noise scheduling strategies namely squared cosine (Nichol & Dhariwal, 2021) and GeoDiff sigmoid (Hoogeboom et al., 2022). In Table 5, we can see that using a completely different noise scheduling strategy at inference actually hampers that model performance. With both Squared cosine and GeoDiff sigmoid denoising, the performance of the diffusion model becomes worse in all three metrics of precision, recall and PQ as compared to the scaler linear denoising (same strategy as in training).

Table 5: Comparison of precision, recall and panoptic quality(PQ) on COCO validation dataset using different inference noise scheduling strategies.

| Scheduling Method | Precision | Recall | PQ |
|---|---|---|---|
| Scaled Linear ($\lambda = 1.0$) | 76.1 | **54.2** | 51.9 |
| Squared Cosine | 16.2 | 19.8 | 13.1 |
| GeoDiff Sigmoid | 75.5 | 52.9 | 51.0 |
| Scaled Linear ($\lambda = 1.3$) | **78.9** | 53.8 | **52.7** |

Figure 4 shows how the Squared cosine (Nichol & Dhariwal, 2021) and GeoDiff sigmoid (Hoogeboom et al., 2022) schedules deviate from the original scaled linear schedule. Unlike the proposed scaling method, the two new schedules significantly deviate towards the start and end timesteps, violating the conditions defined by Kingma et al. (2021). There is also deviation in the middle. The deviation makes the the inference framework move far away from the training conditions resulting in performance loss. We can see the Squared cosine schedule deviates more from the baseline than the Geodiff sigmoid schedule and hence has a significantly lower performance in table 5.

## 6 CONCLUSION

We show that the training noise schedule in diffusion models may be suboptimal for inference. By modifying the noise schedule at inference time, our method increases precision while maintaining recall for panoptic segmentation, and also leads to better text-guided image generation. Our approach allows task-specific tuning and boosts performance without any additional training. It offers a practical deployment strategy and opens up new directions for improving diffusion based models through inference time noise scheduling.

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

# A  APPENDIX

## A.1  PANOPTIC SEGMENTATION ON UNSEEN ADE20K DATASET

To validate the robustness and sensitivity of the modified noise schedule for the task of Panoptic Segmentation, we conducted tests on ADE20K (Zhou et al., 2019) dataset. Here, we used the baseline LDMSeg (Van Gansbeke & De Brabandere, 2024) pre-trained model that was trained on COCO dataset. Since LDMSeg can predict a maximum of 128 instances so we restricted the number of instances in ADE20K dataset samples to 128 (labels in ascending order) and ignored rest of the instances. Note that we do this test to see whether the optimized noise schedule also improve the performance on unseen data.

Table 6: Comparison of precision, recall and panoptic quality on ADE20K dataset with the original training noise schedule $\lambda = 1.00$ and the modified optimum schedule $\lambda = 1.30$. Here also we can see that we get higher panoptic quality with the optimized schedule.

| $\lambda$ | Precision | Recall | PQ |
|---|---|---|---|
| 1.00 | 67.92 | **36.84** | 38.24 |
| 1.30 | **70.82** | 36.37 | **38.61** |

The results are presented in Table 6. Here also, we obtain higher precision of 70.82 and panoptic quality of 38.61 with the modified inference time noise schedule compared to the baseline precision of 67.92 and panoptic quality of 38.24 obtained using the training noise schedule. This shows the robustness of our approach that results in panoptic quality and precision improvements for different datasets. In addition to our experiments with different levels of haze that are presented in the main paper, these results validate that performance improvements hold in diverse settings.

## A.2  SENSITIVITY OF PROPOSED SCHEDULE WITH DEGRADATION

To further evaluate the sensitivity of our method to test-time degradation, we conduct an experiment comparing panoptic segmentation masks generated with and without visual degradation. We assess performance under two different denoising schedules: the original and the modified one. Table 7 presents results on the COCO validation set and its simulated hazy version ($\beta = 0.10$) (Agarwal et al., 2025), evaluated using mIoU and Dice coefficient.

With the original schedule ($\lambda = 1.0$), we observe an average mIoU of 0.926 and a Dice coefficient of 0.962. Under the modified schedule ($\lambda = 1.3$), both metrics improve to 0.948 and 0.973, respectively. This demonstrates that the modified denoising schedule yields segmentation results that are

Table 7: The table shows mean Intersection over Union (mIoU) and Dice coefficient, used to compare the masks obtained from original and degraded images (with haze, $\beta = 0.10$) on COCO validation dataset at different inference noise schedules. The modified schedule achieves higher mIoU and Dice coefficients, indicating lower sensitivity to image degradation.

| Schedule | $\lambda = 1.0$ | $\lambda = 1.3$ |
|---|---|---|
| mIoU | 0.926 | **0.948** |
| Dice | 0.962 | **0.973** |

more consistent across degraded and non-degraded conditions. This shows that denoising with the modified schedule is less sensitive to the degradation as compared to the original schedule. These results prove that modifying the noise schedule do not compromise the robustness of the diffusion model.

