# OpenReview forum: "Noise as a Knob: An Inference Time Noise Scheduling Strategy to Optimize Diffusion Based Vision Tasks"
_ICLR.cc/2026/Conference — ICLR 2026 Conference Withdrawn Submission_

### Official Review · Reviewer_vEXf · 2025-10-27

**Soundness:** 2
**Presentation:** 2
**Contribution:** 2
**Rating:** 2
**Confidence:** 3

**Summary:**

The authors propose an inference-time adjustable noise scheduling strategy, which modifies the noise ratio λ during the reverse denoising process of diffusion models to improve **Panoptic Quality** without retraining. Experimental results demonstrate that the method enhances performance on both panoptic segmentation and text-to-image generation tasks.

**Strengths:**

- The method is entirely based on inference-stage modification, achieving performance improvement without retraining the model, with minimal computational overhead and strong practicality.
- It is validated on both discriminative tasks (panoptic segmentation) and generative tasks (text-to-image generation), showing consistent results and demonstrating that the proposed inference-time scheduling strategy can generalize across different modalities.

**Weaknesses:**

In the panoptic segmentation task, the PQ increases by only about 1.5%, from 51.9 to 52.7, which is a relatively small improvement and still significantly lags behind Transformer-based models, like Mask DINO, which achieves a PQ of 59.5. In fact, according to your experiments, increasing λ over the baseline leads to a decrease in recall, while the improvement in precision is not substantial.

**Questions:**

- In Section 4 Experimental Setup, you state that “We do not conduct any kind of training or fine-tuning,” which implies that inference parameters such as sampling steps and random seeds are fixed. However, under different settings, can the same performance improvement still be maintained? Will the PQ change with the variation of λ?
- You mention that “Our strategy significantly enhances **precision** with minimal **recall**,” but in Tables 2 and 3, only PQ values are reported, without providing **precision** and **recall**.

---

### Official Review · Reviewer_GTqU · 2025-10-29

**Soundness:** 1
**Presentation:** 2
**Contribution:** 1
**Rating:** 2
**Confidence:** 4

**Summary:**

This paper proposes modifying the inference-time noise schedule to improve the performance of diffusion models in downstream tasks, specifically panoptic segmentation and text-to-image generation. The core idea, which involves utilizing a different noise schedule during inference, is adapted from established practices in the image generation literature [2, 3]. Experiments on the LDMSeg model show that the proposed schedule improves segmentation precision at the cost of recall. For text-to-image generation, the method demonstrates a marginal improvement on the X-IQE benchmark.

**Strengths:**

The paper is well-written and clearly organized. The core method is presented in a fluent and easy-to-follow manner, making the paper accessible.

**Weaknesses:**

Weaknesses

1. Disconnect Between Stated Motivation and Proposed Solution.
The paper's core motivation is undermined by a logical inconsistency. The authors attribute the low-precision problem in segmentation to the "inherent stochasticity" of the reverse process (lines 63-65, 149-150). However, the proposed solution—modifying the noise schedule—primarily alters the signal-to-noise ratio (SNR) trajectory during sampling. It does not directly address the source of stochasticity, which stems from the random component in SDE-based samplers. A more direct test of the "stochasticity" hypothesis would have been to compare SDE samplers with their deterministic ODE counterparts. The current mismatch between the identified problem (stochasticity) and the proposed solution (schedule modification) makes the paper's rationale feel unsubstantiated.

2. Limited Novelty and Missing Context.
The experimental setup raises significant concerns about whether the proposed method is a general principle or merely a model-specific fix. The experiments are conducted on LDMSeg, a model based on Stable Diffusion 1.5. It is well-documented that early SD models suffer from a flawed training noise schedule with a non-zero terminal SNR, leading to exposure bias [1, 3]. The low-precision issue observed by the authors could very well be a direct symptom of this known training artifact.
Consequently, the proposed inference-time schedule may simply be a partial remedy for this specific, known flaw in SD 1.5. **This interpretation is not investigated, making the contribution feel more like a targeted workaround for LDMSeg than a generalizable principle.** To demonstrate broader applicability, the authors should have evaluated their method on diffusion models trained with modern, well-posed noise schedules (e.g., v-prediction or zero-terminal-SNR schedules).

3. Limited Novelty and Insufficient Positioning Against Prior Work.
The core technical contribution—decoupling the training and inference noise schedules—lacks novelty. This concept has been extensively explored in foundational diffusion model literature, most notably by Karras et al. [2], which provides a comprehensive analysis of the design space for samplers and schedules. The paper fails to articulate what new insights it offers beyond applying this established technique to a new task (segmentation).
This issue is compounded by the critical omission of a "Related Work" section. Without it, the paper fails to properly contextualize its contribution and differentiate itself from highly relevant prior art. Furthermore, crucial background on the LDMSeg baseline is missing, including its original inference noise schedule, which is the very thing this paper aims to improve upon. This makes it difficult for the reader to assess the true impact of the proposed change.

4. Insufficient Empirical Validation.
The paper's claim of "wide applicability" (lines 85-86) is not substantiated by the narrow set of experiments. The proposed method is only validated on a single segmentation model (LDMSeg) and a single text-to-image benchmark (X-IQE). To support such a broad claim, the empirical evaluation should be expanded to include other diffusion-based segmentation architectures and other standard text-to-image generation benchmarks (e.g., GenEval, DPGBench).

**Questions:**

1. **On the Core Rationale (re: Weakness 1):** To substantiate your claim that "stochasticity" is the primary cause of low precision, please provide a direct ablation study comparing SDE-based samplers with their deterministic ODE counterparts. Does the low-precision issue persist under deterministic ODE sampling? If it does, this would suggest the cause is not stochasticity, prompting a deeper investigation into the true underlying mechanism. Furthermore, can you clarify the conceptual link between modifying the SNR schedule and mitigating sampling stochasticity, as these are generally considered distinct components of the diffusion process?

2. **On Generalizability and Baseline Flaws (re: Weakness 2):** Could the observed performance gains simply be a correction for the known training flaws in the SD 1.5 backbone (i.e., non-zero terminal SNR) [1, 3]? To demonstrate that your method is a generalizable principle rather than a model-specific fix, please provide results of your method applied to a diffusion model trained with a modern, well-posed schedule (e.g., a v-prediction model or one with a zero-terminal-SNR schedule).

3. **On Novelty and Contextualization (re: Weakness 3):**
- In light of foundational works like Karras et al. [2] that extensively explore the design space of inference schedules, what's the primary novel insight of this paper beyond applying an established technique to a new application?
- Please add a dedicated Related Work section to properly situate your method within the existing literature. Besides, please explicitly state the original inference noise schedule used by the baseline LDMSeg model, as this is the critical point of comparison for your proposed method.

4. **On Empirical Validation (re: Weakness 4):** To substantiate the claim of "wide applicability," the empirical scope needs to be broadened. Could you please provide results on:
- At least one other diffusion-based segmentation model to demonstrate portability within the primary task.
- Additional standard text-to-image benchmarks such as GenEval or DPGBench to provide a more robust evaluation of image-text alignment.

In its current state, the paper's contribution appears incremental due to the lack of a deep, principled analysis and its significant overlap with prior work in diffusion model sampling techniques. The empirical evidence is also not strong enough to support its claims.

References
[1] Common Diffusion Noise Schedules and Sample Steps are Flawed.
[2] Elucidating the Design Space of Diffusion-Based Generative Models.
[3] Alleviating Exposure Bias in Diffusion Models through Sampling with Shifted Time Steps.

---

### Official Review · Reviewer_AL8A · 2025-10-30

**Soundness:** 3
**Presentation:** 3
**Contribution:** 2
**Rating:** 2
**Confidence:** 5

**Summary:**

This paper identifies a key limitation of diffusion models in discriminative vision tasks like segmentation: they tend to prioritize recall at the expense of precision, leading to performance gaps compared to transformer-based architectures. To address this, the authors propose a novel, training-free strategy that modifies the noise schedule exclusively during inference. This is achieved by scaling the SNR term $\overline{\alpha}_{t}$ by a power $\lambda$. When $\lambda > 1$, the SNR is effectively lowered, forcing the model to rely more heavily on its condition-aware noise prediction.

**Strengths:**

1. Training-Free Optimization: The most significant contribution is that the method requires zero retraining or fine-tuning. This makes it an extremely practical and computationally efficient tool for optimizing large, pre-trained diffusion models, saving immense time and resources.
2. Broad Applicability and Generalization: The paper successfully demonstrates the strategy's versatility by applying it to two fundamentally different vision paradigms: a discriminative task (panoptic segmentation) and a generative task (text-to-image generation). The consistent positive results in both domains suggest the principle is a general one for conditional diffusion models.

**Weaknesses:**

1. Performance Gap with SOTA Remains Large:  the final PQ of 52.7 still lags significantly behind state-of-the-art specialist models like Mask DINO (59.4 PQ)
2. Manual Parameter Tuning: The optimal $\lambda$ (1.3 for segmentation, 1.2 for T2I) is found empirically through a grid search. The paper does not propose an automatic or adaptive method to determine the optimal $\lambda$. Also, there is a lack of evidence explaining why $\lambda$ was modulated in this way instead of another.
3. Limited Theoretical Grounding: The analogy to CFG is not convincing.
4. Lack of discussion with many related works that discuss the noise schedule in diffusion models. For example, "Improved Noise Schedule for Diffusion Training" surveys different noise schedules for task decomposition, "on the importance of noise scheduling for diffusion models" discover the noise schedule need to be changed for different resolutions, and many other tasks also need to be analyzed.

**Questions:**

Refer to the weakness part.

---

### Official Review · Reviewer_RTYh · 2025-11-05

**Soundness:** 2
**Presentation:** 3
**Contribution:** 2
**Rating:** 4
**Confidence:** 4

**Summary:**

This paper proposes an inference-time modification to diffusion models that improves precision in dense prediction tasks. The key observation is that diffusion-based panoptic segmentation exhibits high recall but low precision compared to transformer-based methods. To address this, the authors modify the reverse diffusion process by replacing ᾱ_t with ᾱ_t^λ (Eq. 6), which reduces the signal-to-noise ratio and forces the model to rely more on conditional predictions. Without any retraining, this achieves PQ 52.7 on COCO (vs. 51.9 baseline) and improves text-image alignment in Stable Diffusion 1.5 from 3.6 to 4.6 on DrawBench. The method is positioned as a flexible "knob" for trading off precision and recall at inference time.

**Strengths:**

1. **Clear problem identification**: The observation that diffusion models prioritize recall over precision (Table 1: 76.1 precision vs 54.2 recall for LDMSeg, compared to 80.4 precision for Mask2Former) is well-articulated and provides strong motivation.
2. **Training-free simplicity**: The method requires only a single hyperparameter λ and no model retraining, making it practical for deployment. This is a significant advantage over methods requiring architectural changes or fine-tuning.
3. **Theoretical grounding**: The SNR analysis (Eq. 7-10) correctly shows that λ>1 reduces SNR, providing mathematical intuition for why the method works. The connection to Kingma et al. (2021)'s finding about endpoint preservation justifies the power transformation form.
4. **Cross-domain validation**: Testing on both discriminative (panoptic segmentation) and generative (text-to-image) tasks demonstrates some generality. The text-to-image results (Table 4: alignment 3.6→4.6) provide orthogonal evidence that the method strengthens conditioning.
5. **Comprehensive λ sweep**: Table 1's ablation across λ∈[0.8, 1.5] clearly shows the precision-recall trade-off, and the finding that λ<1 hurts both metrics validates the design choice.

**Weaknesses:**

### W1: Insufficient theoretical justification for design choices

The paper provides no derivation for why the power transformation ᾱ_t^λ is optimal among transformations that reduce SNR while preserving endpoints. Alternative forms like ᾱ_t' = (1-β)ᾱ_t or exponential decay are never explored. Table 5's comparison with Squared Cosine and GeoDiff Sigmoid is insufficient since these violate endpoint preservation—comparing against other endpoint-preserving forms would validate the power form's uniqueness. More critically, the causal link from "lower SNR → higher precision" is never established. Why does stronger conditioning reduce false positives specifically rather than affecting recall equally?

**Actionable suggestion**: Add ablation comparing ᾱ^λ against other endpoint-preserving transformations. Decompose Table 1 into TP/FP/FN changes to reveal the mechanistic pathway from λ to precision improvement.

### W2: Severely limited experimental scope

Despite claiming broad applicability and being training-free, the paper tests only 2 models. For discriminative tasks, only LDMSeg is evaluated despite citing Tan et al. (2022) Semantic Diffusion Network, Qiu et al. (2024) AlignDiff, and Chen et al. (2023a) DiffusionDet—none are tested. This leaves unclear whether precision-recall imbalance is a general diffusion property or LDMSeg-specific. For generative tasks, only SD 1.5 (2022) is tested, ignoring modern models like SD 2.1 or SDXL. Given training-free nature and modest compute costs, this limited scope is unjustifiable.

**Actionable suggestion**: Test on Semantic Diffusion Network (semantic segmentation), DiffusionDet (object detection), and SD 2.1/SDXL (text-to-image) to demonstrate generality across ≥5 models rather than 2.

### W3: λ sensitivity and deployment concerns

Optimal λ varies significantly across conditions: λ=1.3 for clean images, λ=1.1 for heavy haze (Table 3), and improvements shrink on unseen data (COCO +0.8 vs ADE20K +0.37 PQ in Table 6). This suggests λ requires per-dataset validation tuning, yet no automatic selection method is provided. Positioning λ as a flexible "knob" is misleading without guidance for practical deployment.

**Actionable suggestion**: Propose a heuristic for automatic λ selection (e.g., based on image quality metrics) or demonstrate that a single λ value works adequately across diverse settings.

### W4: Missing critical baselines

Section 3.2 claims the method is "analogous to Classifier-Free Guidance" but never compares against CFG as a baseline. If truly analogous, CFG (with appropriate unconditional baseline definition for segmentation) should achieve similar results—its absence raises questions about the necessity of the proposed approach. Other inference-time methods (guidance in ControlNet, iterative refinement) are also not compared.

**Actionable suggestion**: Attempt CFG adaptation to panoptic segmentation or explain why it's fundamentally incompatible, which would justify why a new method is needed.

### W5: Lack of mechanistic analysis

While Table 1 shows λ↑ causes precision↑ and recall↓, the underlying mechanism is unexplored. Is precision gain from fewer spurious regions (FP↓), more conservative thresholding, or sharper boundaries? Figure 2 qualitatively suggests reduced over-segmentation, but this is never quantified. Without TP/FP/FN decomposition, the causal pathway from SNR manipulation to performance change remains opaque.

**Actionable suggestion**: Add analysis showing how TP/FP/FN counts change with λ, and plot IoU distributions to test whether the method improves quality or simply becomes more conservative.

**Questions:**

**Q1**: Can you provide results on Semantic Diffusion Network (Tan et al. 2022) and DiffusionDet (Chen et al. 2023a)? If the method fails on these models, it would suggest LDMSeg-specific effects rather than general diffusion properties.

**Q2**: Have you compared alternative transformation forms (e.g., ᾱ' = aᾱ + b with a<1) that also reduce SNR? If yes, what were the results? If no, can you add this ablation?

**Q3**: What happens if λ is made timestep-dependent, e.g., λ(t) = 1 + γ·sin(πt/T) to focus modification on middle timesteps per Kingma et al. (2021)? Does this improve performance or robustness?

**Q4**: Can you provide TP/FP/FN breakdowns for Table 1? Specifically, as λ increases from 1.0→1.5, do FP counts decrease monotonically?

**Q5**: Why doesn't the paper compare against CFG as a baseline? Can CFG be adapted to panoptic segmentation, and if so, how does it compare to your method?

**Q6**: For ADE20K results (Table 6), the improvement is much smaller (+0.37 vs +0.8 on COCO). Is this because λ=1.3 is COCO-specific? What is the optimal λ for ADE20K if you sweep it separately?

**Q7**: Table 5 shows that alternative schedules (Squared Cosine, GeoDiff Sigmoid) fail dramatically. But these violate endpoint preservation. Have you tested schedules that preserve endpoints but use different functional forms than power (e.g., piecewise linear)?

---

### Note · Authors · 2025-11-21

**Comment:**

We thank all the reviewers for their constructive feedback. We believe that our article needs substantial experiments and we will not be able to address all the reviewer comments during this rebuttal period. We would therefore like to withdraw our article at this point and will resubmit it after addressing the feedback.

**Withdrawal Confirmation:**

I have read and agree with the venue's withdrawal policy on behalf of myself and my co-authors.